# Recent Advances in the Development of Virus-Like Particle-Based Flavivirus Vaccines

**DOI:** 10.3390/vaccines8030481

**Published:** 2020-08-27

**Authors:** Naru Zhang, Chaoqun Li, Shibo Jiang, Lanying Du

**Affiliations:** 1Department of Clinical Medicine, School of Medicine, Zhejiang University City College, Hangzhou 310015, China; zhangnr@zucc.edu.cn (N.Z.); lcqlcq2019@outlook.com (C.L.); 2School of Basic Medical Sciences, Fudan University, Shanghai 200433, China; 3Lindsley F. Kimball Research Institute, New York Blood Center, New York, NY 10065, USA

**Keywords:** flaviviruses, Zika virus, Dengue virus, West Nile virus, Japanese encephalitis virus, virus-like particles, vaccines

## Abstract

Flaviviruses include several medically important viruses, such as Zika virus (ZIKV), Dengue virus (DENV), West Nile virus (WNV) and Japanese encephalitis virus (JEV). They have expanded in geographic distribution and refocused international attention in recent years. Vaccination is one of the most effective public health strategies for combating flavivirus infections. In this review, we summarized virus-like particle (VLP)-based vaccines against the above four mentioned flaviviruses. Potential strategies to improve the efficacy of VLP-based flavivirus vaccines were also illustrated. The applications of flavivirus VLPs as tools for viral detection and antiviral drug screening were finally proposed.

## 1. Introduction

Flaviviruses have posed an increasing threat to public health worldwide due to their potential to cause severe diseases. Flaviviruses are normally transmitted to humans through infected mosquitoes or ticks [1], but some of them have acquired the ability to transmit to people through other routes, such as sex contact or blood [2,3,4,5,6]. The most important human pathogenic flaviviruses include Zika virus (ZIKV), dengue virus (DENV1-4), West Nile virus (WNV), Japanese encephalitis virus (JEV), and yellow fever virus (YFV). These viruses belong to the flavivirus genus in the family of *Flaviviridae* [7]. Among these flaviviruses, DENV1-4, WNV and JEV may cause millions of infections and tens of thousands of deaths each year [8].

Flavivirus virions have a diameter of around 50 nm [9], whose genomes contain a positive-sense, single-stranded RNA with about 11 kilobase pairs (Kb) in length [10]. The genomic RNA consists of an open reading frame (ORF) that is flanked by a 5′ untranslated region (5′-UTR) and a 3′-UTR. The ORF encodes a polyprotein that is post-translationally processed, through the cleavage of viral and cellular proteases, into 10 proteins, including three structural proteins, such as capsid (C), premembrane (prM), and envelope (E) proteins, as well as seven nonstructural (NS) proteins, such as NS1, NS2A, NS2B, NS3, NS4A, NS4B and NS5 [10]. Structural proteins mainly participate in viral particle formation. NS proteins are responsible for viral RNA replication, but they may also play roles in viral assembly [11,12] and evasion of immune responses [13,14].

Flaviviruses are composed of a nucleocapsid (N), whereas the N protein consists of the C protein and a positive-sense, single-stranded RNA. The N protein is surrounded by a lipid bilayer, while the latter is inserted by M and E proteins. The M protein is a precursor protein (prM). The immature particles contain 60 trimeric spikes of prM-E heterodimers (Figure 1A), which are cleaved before mature virus particles, and then released from infected cells [15]. Virus particles become maturated during virus egress through the secretary pathway; furin protease in the Golgi cleaves the prM protein, and releases the pr peptides as the virus reaches neutral pH upon exit from host cells [16]. The mature flavivirus particles consist of 90 E homodimers and 90 M homodimers on the E proteins (Figure 1B). The E proteins are responsible for receptor binding, attachment, viral entry and membrane fusion.

Entry of flaviviruses is mediated by attachment of viral E proteins into host receptor(s) and clathrin-mediated endocytosis. To initiate this process, low pH in the endosomes may trigger viral E proteins to change conformation, mediating the fusion between viral and endosomal membranes, thus leading to the release of the viral genome into the cytoplasm for replication. The immature viral particles are assembled on the surface of endoplasmic reticulum (ER), and then cleaved by furin, followed by the formation of mature viral particles, and released by exocytosis (Figure 2). The E proteins consist of three domains, including domain I (DI), DII and DIII. There is a high-level structural homology among the flavivirus E proteins [17,18,19,20], but their amino acid sequences have about 60% difference [1]. Due to the critical role of flavivirus E proteins in host cell entry and membrane fusion, they are the major targets of vaccines with the ability to induce neutralizing antibodies that mediate protection and long-lasting immunity.

Vaccination is considered as an effective strategy to prevent viral infection. Currently licensed flavivirus vaccines include those targeting YFV, JEV, and DENV [21]; several vaccines against ZIKV have been evaluated in clinical trials [22,23,24]. The development of an effective DENV vaccine is generally challenging because immunogenicity and protection need to be elicited against all four serotypes. In addition, vaccination that induces low-level and cross-reactive antibodies might cause more severe infection, resulting in antibody-dependent enhancement (ADE) [25]. CD4^+^ T cells play an important role in generating effective immune responses by coordinating antibody production and the development of cellular immune memory [1]. Currently developed flavivirus vaccines include live-attenuated [26,27,28], inactivated [22,29], plasmid DNA [30,31,32], viral vectored [33,34,35], recombinant protein [36,37,38], YFV chimera [39], and virus-like particle (VLP)-based vaccines [40]. While some inactivated virus and live-attenuated virus-based vaccines might have potential safety issues, live-attenuated yellow fever 17D vaccine is considered an effective and safe vaccine [28,41]. DNA-based vaccines encode virus-specific genes, and they are convenient for sequence editing; several DNA vaccines such as those targeting ZIKV have been proceeded in clinical trials [24,42]. Viral vectors, such as adenovirus and canarypox, can be used to develop vaccines against flaviviruses [34,35,43]. A concern might be raised regarding pre-existing immunity against the viral vector (such as human adenovirus 5 (Ad5)), which may result in reduced immunity and efficacy of the vaccines.

Development of safe, effective, and economical flavivirus vaccines with easy and cost-effective production for large-scale use is critical. In this regard, VLP-based vaccines may meet these criteria, since they induce effective immune responses, and can be readily manufactured with cost-effective production on a large scale [44]. Moreover, VLPs are safe since they do not contain any viral genetical materials and are thus non-infectious. VLPs may consistently display viral antigens with high density on their surface, having a potential for high antigenicity and potent immunogenicity [45,46,47]. Therefore, VLPs provide a promising approach for developing safe and effective flavivirus vaccines. In this review, we summarize recent advancements in the development of VLP-based flavivirus vaccines (Table 1, Table 2, Table 3 and Table 4).

## 2. VLP-Based Flavivirus Vaccines

VLPs present multi-protein structures mimicking native virions. VLPs have capacity to elicit strong humoral and cellular immune responses, as well as protection, thus serving as excellent platforms for the development of efficient vaccines. Flavivirus VLPs are relatively smaller as compared to the intact virions, and they can be produced in a large variety of expression platforms, including yeast, plant, insect cells, and mammalian cells [48,49,50,51,52]. VLPs have proven to be effective and safe vaccine antigens in preclinical or early-stage clinical studies with ZIKV, DENV, WNV, and JEV [52,53,54,55,56].

### 2.1. VLP-Based Vaccines against ZIKV Infection

ZIKV belongs to the *Flaviviridae* family. ZIKV is a mosquito-borne flavivirus and has caused a severe threat to public health. ZIKV infection may lead to microcephaly and serious neurological complications [57]. The positive-sense viral RNA genome (11 Kb) is translated in the cytoplasm to generate three structural proteins (C, prM/M and E) and seven non-structural proteins [58]. The structural proteins play a key role in virus assembly, predominantly in the ER lumen. ZIKV VLPs can be produced in cells expressing viral E [57] and/or prM-E proteins. For example, Alexander’s group developed a VLP consisting of ZIKV prM and E proteins in transiently transfected HEK293 cells [59]. The noninfectious particles morphologically resemble the live virus, and can induce effective immune responses against ZIKV [40,60].

A number of VLP-based ZIKV vaccines have been evaluated in animal models (Table 1). It has been shown that a ZIKV-VLP expressing E protein domain III (EDIII) CD loop antigen elicited specific immune responses and protection against ZIKV infection [61]. Notably, the addition of the flavivirus C protein may promote virion stability [62], and thus immunogenicity of VLP vaccines. For example, Garg et al. generated ZIKV VLPs using prM-E and C-prM-E constructs, and tested them in mice [63]. The C-prM-E VLP vaccine induced slightly higher antibody responses than the prM-E VLPs. In addition, a Zika subunit VLP vaccine was constructed containing ZS protein (an in-frame fusion of ZIKV DIII with the Hepatitis B virus (HBV) surface antigen) and S protein (an unfused HBV surface antigen) using the methylotrophic yeast *Pichia pastoris* expression system [64]. These VLPs induced antibodies in BALB/c mice, neutralizing against ZIKV reporter virus particles. Importantly, the VLP-induced antibodies did not enhance sublethal DENV-2 challenge in a mouse model deficient in IFN-α/β and IFN-γ receptors (AG129), suggesting it to be a promising vaccine candidate.

VLP-based ZIKV vaccines can be displayed on different carriers. Domain III of ZIKV E protein (zDIII) was assembled into VLPs displayed on the hepatitis B core antigen (HBcAg). Two doses of HBcAg-zDIII VLPs elicited potent humoral and cellular responses in mice and demonstrated protective immunity against the infection of multiple ZIKV stains [65]. Particularly, HBcAg-zDIII VLP-elicited antibodies did not enhance the infection of DENV in Fc gamma receptor-expressing cells [65], suggesting that zDIII-based vaccines eliminated the safety concern. Basu et al. also displayed B cell epitopes (amino acids 241–259, 294–315, 317–327, 346–361, 377–388 and 421–437) of ZIKV E protein on bacteriophage VLPs. They found that VLPs displaying a single B-cell epitope reduced ZIKV infection, while immunization with a VLP mixture displaying combined B-cell epitopes neutralized ZIKV infection [66], suggesting that VLPs displaying multiple ZIKV B-cell epitopes, rather than the VLPs displaying a single B-cell epitope, provide a promising strategy to enhance ZIKV neutralizing activity.

Except for immunization, the passive transfer of a ZIKV VLP vaccine-immune sera may protect the immunocompromised AG129 mice from lethal ZIKV challenge [59], providing the feasibility for passive protection of ZIKV infection.

### 2.2. VLP-Based Vaccines against DENV Infection

#### 2.2.1. VLP-Based DENV Vaccines

Dengue is an emerging public health problem worldwide caused by any one of the four mosquito-borne DENV serotypes with distinct antigenicity. It was first reported in 1997 that expression of the DENV structural proteins C-prM-E in yeast *Pichia pastoris* led to the generation of VLPs [67]. When produced in temperatures as low as 31 °C, VLPs, consisting of the structural proteins C-prM-E, as well as a modified complex of the NS2B/NS3 protease, elicited the highest neutralizing antibodies in mice [68], suggesting that temperature may affect the conformation of E protein.

Since then, VLPs comprised of fewer antigen components have been developed, which can be expressed in a variety of systems and/or induce sufficient immune responses (Table 2). For example, DENV1 VLPs, containing prM and E proteins, were expressed in yeast *Pichia pastoris*, and induced viral neutralizing antibodies and T cell responses in mice [69]. VLPs consisting of DENV4 E ectodomain were developed in *Pichia pastoris*, which were immunogenic in mice and elicited DENV4 specific neutralizing antibodies [70]. The prM and E protein-expressing DENV1-4 VLPs can be efficiently produced in HEK293 cell suspension culture, displaying epitopes on the virus particles [48]. Recombinant VLPs expressing prM and E genes of DENV1-4 serotypes, which were transiently expressed in HEK293T cells, elicited specific humoral and cellular immune responses in mice [71]. DENV E-based VLPs without the prM protein could induce high titers of neutralizing antibodies [72]. DENV2 VLPs were produced from mosquito cells stably expressing prM-E with enhanced prM cleavage. These VLPs elicited high-level neutralizing antibodies in mice, and also induced potent EDIII-specific antibody responses and neutralizing antibodies in macaques primed with live-attenuated virus [73], providing an immunization platform for induction of robust neutralizing antibody responses in protection against DENV-2 infection and the related diseases.

Antibodies to a DENV serotype with suboptimal protective activity against that serotype may enhance infection by other DENV serotypes, leading to ADE. The only licensed dengue vaccine, Dengvaxia, a chimeric tetravalent dengue vaccine containing a live-attenuated YFV strain 17D backbone and prM and E genes of DENV1-4 serotypes, is unable to confer a balanced protection against all the serotypes [39,74]. In addition, because of its low efficacy in children younger than nine years old, this vaccine enhances dengue infection in these individuals, increasing the risk of hospitalization and thus raising safety concerns [39,75,76]. Therefore, various VLP-based approaches have been explored for development of dengue vaccines against multiple DENV serotypes with improved immunogenicity and protection. VLPs expressing three distinct DENV2 E protein epitopes exhibited significant humoral immune responses against individual variations [77]. A tetravalent VLP vaccine expressing prM and E proteins of four DENV-VLP serotypes were successfully expressed in *Pichia pastoris*, and induced higher-titer antibodies compared with individual VLP dengue serotypes [52]. Recombinant DENV1-4 VLPs expressing prM and E proteins with a F108 mutation in the E fusion loop region increased VLP production in a mammalian cell system, and elicited high-level neutralizing antibodies against all four DENV serotypes without the presence of ADE [50]. DENV1, DENV2 and DENV3 VLPs expressing *EDIII* epitopes could elicit virus-specific neutralizing antibodies [78,79,80,81]. A bivalent mosaic VLP vaccine co-expressing E proteins of DENV1 and DENV2 in yeast *Pichia pastoris* maintained specific antigenicity and elicited EDIII-specific neutralizing antibodies in mice [82]. A tetravalent mosaic VLP vaccine co-expressing E proteins of four DENV serotypes in *Pichia pastoris* retained serotype-specific antigenicity and immunogenicity of all four DENV serotypes without induction of ADE in mice [83]. To further increase the immunogenicity of VLPs, the EDI-EII regions of the Cosmopolitan genotype vaccine construct can be replaced by the Asian 1 EDI-II to produce VLPs in elicitation of significantly high IgG antibodies with neutralizing activity [84].

In spite of the potential application of DENV VLPs as promising vaccine candidates, there has been challenging in large-scale production of these particles, in addition to the presence of antibody-enhanced effects described above. The yield of VLPs can be improved by substituting the DENV prM-E expression cassettes of the prM signal peptide and E protein stem-anchor region with homologous cellular and viral counterparts, respectively. Codon optimization and mutation of the E fusion loop regions are able to promote VLP maturation during export [85]. These modification approaches on expression cassettes are expected to generate stably expressing clones, and thus generate DENV VLPs for large-scale use.

#### 2.2.2. Chimeric DENV VLP-Based Vaccines

A variety of chimeric VLP vaccines against DENV have been developed and tested in animal models. A mosaic DSV4 VLP was designed by fusing EDIII regions of four DENV serotypes with HBV surface (S) protein, and induced potent and long-lasting neutralizing antibodies against DENV1-4 infection in mice and macaques. Passive transfer of the immune sera prevented DENV-susceptible AG129 mice from mortality and production of inflammatory cytokines [86]. Another chimeric VLP vaccine encoding DENV EDIII and HBcAg was immunogenic in inducing EDIII-specific antibodies with neutralizing activities [80]. These studies suggest the use of HBV-S and HBcAg as VLP platforms for the display of DENV EDIII proteins. In addition, chimeric DENV VLPs can be generated by fusing the coding gene of Junin virus (Z-JUNV) Z protein with cryptic peptides conserved among E proteins of DENV1-4 serotypes, which induced specific antibody responses with neutralizing activity against DENV infection [87].

Chimeric DENV VLPs can also be generated by displaying a viral EDIII region on HBsAg S and expressing the antigens in measles virus (MV) vector. A MV vectored VLP vaccine expressing DENV2 EDIII-HBsAg S proteins was immunogenic in mice and induced robust neutralizing responses against MV, HBV, and DENV-2 [88]. Moreover, CD16-RIgE, a chimeric human membrane glycoprotein containing CD16 ectodomain and gamma chain of IgE receptor (RIgE), was used as a VLP platform to display DENV1 E protein DIII, in which CD16 ectodomain was replaced with DENV1 EDIII. Co-expression of EDIII-RIgE and HIV-1 Pr55Gag polyprotein precursor (Pr55GagHIV) in insect cells led to the incorporation of VLPs with EDIII displayed on the surface [89]. This EDIII-expressing chimeric VLPs induced EDIII-specific neutralizing antibodies against DENV in mice, suggesting the potential for using CD16-RIgE as a platform to develop flavivirus VLP vaccines.

#### 2.2.3. Factors Affecting the Production or Packaging of DENV VLPs

DENV E protein plays an important role in mediating viral entry and assembly of progeny virus during cellular infection. Site mutations of the conserved residues in the DI/DIII region of DENV-2 E protein lead to mutations that affect virus assembly [90]. Except for DENV-2 VLPs, other recombinant flavivirus VLPs can be efficiently packaged by co-expression of viral prM and E proteins. Part of the reasons in inefficient production of such VLPs might be due to the stem-anchor of DENV-2, which contains an ER retention signal. Notably, DENV-2 VLP production could be enhanced by the co-expression of C protein [91], suggesting some potential mechanisms that facilitate viral particle formation during DENV-2 replication.

Since DENV VLPs do not contain viral genome, they are much smaller than the wild-type virus particles. However, the VLPs do maintain the antigenic properties of wild-type virus particles. Importantly, antibody epitopes with good conformation and quaternary structure of virus particles can be efficiently displayed on the surface of DENV1-4 VLPs [48], supporting the VLPs as an effective approach to develop safe and efficient DENV vaccines.

### 2.3. VLP-Based Vaccines against WNV Infection

#### 2.3.1. VLP-Based WNV Vaccines

VLPs targeting WNV antigens have been designed and tested in animal models (Table 3). VLPs expressing WNV structural proteins prME or CprME were produced in an insect cell expression system, and induced WNV-specific antibodies in mice with potent neutralizing activity. In addition, immunization with prME-VLPs induced sterilizing immunity without viremia or viral RNAs in the spleen or brain, and no evidence of morbidity or mortality after WNV challenge [92]. Two WNV VLPs have been developed through sediment analysis in sucrose density gradients [93]. One is fast-sediment VLPs (F-VLPs), consisting of M and E proteins with the particle size of 40–50 nm, and the other is slow sediment VLPs (S-VLPs), consisting of prM and E proteins with the particle size of 20–30 nm. F-VLPs elicited higher-titer WNV-specific IgG and neutralizing antibodies than S-VLPs, and also had higher protective efficacy than S-VLPs against WNV infection in mice, suggesting that F-VLPs mimic the virions better than S-VLPs, thus eliciting a better immune response than S-VLPs [93].

The immunogenicity of VLP-based WNV vaccines can be improved by conjugating the EDIII fragment to VLPs of bacteriophage AP205. A WNV vaccine generated by chemically crosslinking viral EDIII to VLPs derived from bacteriophage AP205 elicited high titers of WNV-neutralizing antibodies, completely protecting mice from WNV infection [94]. AP205-DIII mosaic VLPs were generated by displaying a 111-amino acid WNV EDIII on the surface of AP205 VLPs, and expressing them in the *E.coli* system. These VLPs stimulated anti-DIII responses in mice with high-level IgG2 isotype antibodies [95], suggesting the possibility of development of AP205-DIII VLPs as a promising West Nile vaccine. Similar to DENV VLPs, CD16-RIgE can be used as a platform to display WNV EDIII on the surface of VLPs, which induced neutralizing antibodies in mice [89].

#### 2.3.2. Chimeric WNV VLP-Based Vaccine

A few chimeric VLP vaccines targeting WNV have been developed. For example, WNV prM, E, and C proteins were fused into herpes simplex virus 1 (HSV-1) d106 viral vector for the extracellular production of VLPs, which elicited WNV-specific IgG antibody response in immunized mice [55].

#### 2.3.3. Factors Affecting the Production or Packaging of WNV VLPs

This may be affected by various factors. Expression of WNV E protein without prM led to the proteolytic cleavage of the expressed E protein [55]. Co-expression of three WNV structural proteins in an insect cell expression system resulted in the assembly of VLPs in high yields [96]. Mutations in the prM protein (T20D, K31A, K31V, or K31T) reduced, or eliminated, the secretion of WNV VLPs, whereas they had less effect on wild-type WNV [97]. Different substitutions at specific fusion peptide residues of WNV E protein also affect antibody reactivity and VLP secretion [98]; part of the reasons might be due to the differences in the quantity of prM protein, or in the maturation, structure, and symmetry of WNV VLPs. In the absence of viral protease, the number of amino acid residues consisting of the N-terminal prM-E proteins influences the efficiency of VLP assembly, maturation, and release [99].

### 2.4. VLP-Based Vaccines against JEV Infection

#### 2.4.1. VLP-Based JEV Vaccines

Japanese encephalitis (JE) is a vector-borne encephalitis disease caused by JEV. This disease distributes extensively, particularly in Asian and Western Pacific countries, in addition to northern Australia [100]. JEV leads to fatality rates of about 20–30%, with neurologic or psychiatric sequelae being observed in 30–50% survivors [101]. No specific treatment has been available for the prevention of JE disease, and vaccination is a practical and effective approach to prevent JEV infection in humans and domestic animals [102].

VLP-based JEV vaccines can be produced in different expression systems with immunogenicity in animals (Table 4). A stable and high-producing cell clone, J12#26, which possessed hemagglutinating activity, as well as JEV E and M proteins, was adapted to serum-free medium to produce VLPs [103]. A JEV VLP containing prM and E proteins was developed in a *Pichia pastoris* expression system and showed efficient immunological properties in mice [104]. Expression systems using lepidopteran insect cells, such as Sf9 and High Five insect cells, can be employed for the efficient production of JE VLPs. JEV VLPs, containing prM and E glycoprotein, were produced in stably transformed High Five cells and the yield of E protein was higher than that in the baculovirus-insect cell system [105]. A secreted form of JEV VLPs has been produced using Sf9 insect cells following infection with a recombinant baculovirus that contains the JEV authentic prM gene and E gene downstream of the polyhedron promoter [106]. VLPs produced by using the same expression system were evaluated for their protective efficiency in mice, and they conferred complete protection against JEV-P3 strain [107].

JEV vaccines with or without adjuvants may induce desirable immune responses and/or protection. In the absence of adjuvants, the VLPs produced in cell clone J12#26 induced high neutralizing antibody titers in mice and confer complete protection against wild-type JE virus challenge [103,108], suggesting that the recombinant E-VLP antigen produced by this J12#26 cell clone is a promising second-generation JE vaccine. In the presence or absence of an adjuvant, a secreted form of JE VLPs containing JEV M and E proteins induced high titers of neutralizing antibodies and 100% protection against lethal JEV challenge in mice [102]. When mice were immunized with VLPs formulated in aluminum (Alum) adjuvant, sera obtained relatively high neutralizing titers [93]. A JE-VLP containing JEV E protein was used to immunize mice with either poly (γ-glutamic acid) nanoparticles or Alum adjuvant, achieving long-lasting protection in 90% of the mice [109].

VLP-based JEV vaccines are able to induce protection against infection of divergent JEV strains. Nerome et al. developed a Nakayama JE VLP in silkworms using codon-optimized prM and E genes, and the VLPs showed protection against both homologous and heterologous strains (Beijing-1 and Muar) [110]. The same group successfully generated a Muar JE VLP vaccine in silkworm pupae through codon optimization of E gene [111]. A genotype I JE VLP, consisting of prM and E proteins, was used to establish a mouse model exhibiting neurological symptoms, and it showed potent protection against homologous or heterologous JEVs [112]. Moreover, VLP-immunized swine presented neither fever nor viremia after challenge with genotype I or III JEV, suggesting that these VLPs could provide sterile protection against Genotype I and III JEV infection in swine [112].

JEV vaccines presented on different carriers have been revealed. For example, four peptides from various locations within JEV E protein were fused to Johnson grass mosaic virus (JGMV) coat protein and auto-assembled to generate VLPs. Immunization of mice with the recombinant VLPs containing JEV peptide sequences induced anti-peptide and anti-JEV antibodies with neutralizing activity, which conferred protection against lethal JEV challenge [113].

#### 2.4.2. Chimeric or Vectored JEV VLPs

Chimeric JEV VLPs can be generated by inserting the prM and E genes of a genotype 3 JEV strain into lentiviral TRIP vector. Mice and pigs immunized with the lentiviral vector expressing JEV VLPs produced JEV-specific IgG antibodies and neutralizing antibodies against genotypes 1, 3, and 5 of JEV strains, which were sufficient to confer protection of immunized pigs against different genotypes of JEV [114]. These studies suggest the potential of using lentiviral vector expressing JEV VLPs as a vaccine candidate in endemic regions where more than one genotype of JEV is circulating. The chimeric VLPs not only can be used as promising vaccine candidates, but also they can be applied to study viral genome packaging and cellular factors involved in vector specificity in terms of the safety profiles [115].

#### 2.4.3. Factors Affecting JEV VLP Production

Both prM and E proteins of JEV contain a single N-glycosylation site, and the role of N-glycosylation of these proteins may affect protein folding and activity. It has been shown that the single N-liked glycosylation site in the JEV prM protein is important for virus particle release, whereas removal of the N-glycosylation site from the prM and/or E protein of JEV resulted in low efficiency of protein folding and thus affected VLP formation, as well as secretion [116,117]. Moreover, different expression systems may affect the generation of glycosylated VLPs. Compared to the insect cell and *E.coli* expression systems, mammalian and yeast cell expression systems tend to form VLPs with good conformation and correct folding with appropriate glycosylation, thus they are applicable for expression of effective VLPs.

## 3. Prospects and Challenge of VLP-Based Flavivirus Vaccines

As described above, VLP-based vaccines have been demonstrated to induce protective efficacy against flavivirus infection in animal models. Notably, multiple injections and/or strong adjuvants might be required to reach sufficient efficacy, highlighting the need for more immunogenic and safe VLP-based flavivirus vaccines. For example, JEV VLPs without an adjuvant only protected 50% of the mice, whereas JEV VLPs with either poly (γ-glutamic acid) nanoparticles or Alum adjuvant achieved protection in 90% of the mice, and the protection was long-lasting [109]. In baculovirus expression system, when lepidopteran insect cells, such as Sf9 and High Five cells, are used as host cells for stably expressing VLPs, the choice of a prompter to drive the heterologous gene is crucial, since a weak promoter might result in low yields of recombinant proteins. VLPs can be purified by different methods, such as diafiltration, sucrose density centrifugation and cellufine sulfate column chromatography. However, different purification methods may affect the immunogenicity of the VLPs. With the same purification method, different sedimentation speeds might affect the size and immunogenicity of WNV VLPs [93].

ADE is a phenomenon in which pre-existing antibodies with no or low-titer neutralizing activity result in enhanced viral infection. Specifically, the Fc region of the non-neutralizing antibody binds to the fragment crystallizable gamma receptor, leading to the subsequent C1q interactions and immune effector functions, and thus the ADE. There is a serious concern on ADE associated with antibodies to flaviviruses, such as DENV and ZIKV. It has been reported that DENV antibodies from the sera of people living in a DENV-endemic area are capable of enhancing ZIKV infection in a human macrophage-derived cell line and primary human macrophages [118]. Therefore, when designing flavivirus vaccines, the ADE effect should be considered, and strategies to avoid ADE are essential. For example, to eliminate the ADE caused by VLP-based DENV vaccines, bivalent mosaic VLPs were generated by co-expressing the E proteins of DENV-1 and DENV-2 in a *Phichia pastoris* expression system, which induced EDIII-specific neutralizing antibodies lacking the ADE [82].

## 4. Other Potential Applications of Flavivirus VLPs

In addition to serving as vaccines against flavivirus infections, VLPs can also be used diagnostically to detect flaviviruses or facilitate drug discovery.

### 4.1. Diagnostic Assays of VLPs for Flavivirus Detection

Millions of people have close contact with flavivirus-carrying mosquitos every year, resulting in thousands of deaths. Cocirculation of flaviviruses constituting similar genetic backgrounds—such as JEV, WNV, DENV, ZIKV, or YFV—in the same areas leads to the generation of cross-reactive antibodies, which raises concerns for the development of effective vaccines, as well as diagnostic tests [56]. Generally, virus-infected tissue cultures or animal tissues can be applied as viral antigens for diagnostic tests. VLPs can also be used as a diagnostic tool by their antigenic properties in the recognition of native virions. Co-expression of C, M and E structural proteins of WNV yielded noninfectious WNN VLPs, and the VLPs were shown to perform correctly in two different ELISAs for WNV diagnosis [96]. The potential use of VLPs as diagnostic tests for the above-mentioned common flaviviruses has also been reported [56]. In order to make a reliable source of standardized viral antigens for serodiagnosis of medically important flaviviruses, eukaryotic plasmid vectors expressing prM/M and E proteins, which self-assemble into noninfectious VLPs of different flaviviruses, have been developed. Serum samples from patients infected with Powassan virus or La Crosse encephalitis virus have been used for evaluation of the cross-reactivity of VLP antigens. The results from MAC-ELISA and receiver operating characteristic (ROC) curve analyses demonstrated that VLP antigens presented higher sensitivity and specificity compared to the traditional assays using samples collected from virus-infected tissue culture or suckling mouse brain tissues [119].

### 4.2. Flavivirus VLPs Can Be Used as Tools for Antiviral Drug Screening

The development of flavivirus chemotherapy requires high-throughput screening (HTS) assays. DENV VLPs with high production and infectivity were produced by either electroporating replicon RNA into a stable cell line expressing the structural proteins or by the sequential electroporation of the replicon RNAs and the structural gene RNAs. VLP infection with cell-based HTS assay provides a safe and rapid approach for screening inhibitors against different life cycles of DENV [120]. Three cell-based HTS assays were compared for WNV drug discovery, and one of the assays used a cell line harboring a persistently replicating sub-genomic replicon. The second assay is based on a full-length reporting virus, and the third used packaged VLPs containing replicon RNA. Because each assay encompassed multiple, but discrete, steps of the viral life cycle, the three assays could potentially be used together to discriminate the mechanism of action for any inhibitor of viral entry, replication and virus assembly [121], suggesting that the cell-based antiviral HTS approach can be developed for other flaviviruses. The phenomenon of syncytial formation is common among flaviviruses induced by the E protein of VLPs in insect cells. ZIKV VLPs generated by E protein-induced ZIKV-E-specific polyclonal antibodies, which significantly reduced syncytial formation in insect cells [57]. This result suggests that the inhibitory effect on syncytial formation can be developed as a novel antiviral screening approach for inhibitory antibodies, peptides, or small molecules targeting the E protein of other flaviviruses.

## 5. Conclusions

In this review, we summarized VLP-based vaccines for several medically important flaviviruses. A number of points still need to be addressed in order to improve the efficacy of vaccine candidates. First, antigen choice should be considered. Although C-prM-E is reported to be highly immunogenic, ADE will not be avoided by complete E protein. Safety is a crucial consideration when designing VLP vaccines. Second, administration routes and choice of appropriate adjuvants need to be taken into consideration, which could be facilitated by selecting appropriate animal models for pre-clinical studies. Finally, clinical trials should be carefully designed and planned to verify the feasibility of potential vaccine candidates. Nevertheless, results from animal studies may not be directly extrapolated and interpreted into humans, even though pathological pathways could be similar between animals and humans. Therefore, data from animal models should be carefully reviewed and evaluated to determine the impact of a vaccine on pathological pathways before moving the pre-clinical vaccines into human clinical trials. Overall, VLP vaccines for flavivirus should be further studied to develop more safe and effective vaccine candidates, in order to protect humans against potential flavivirus outbreaks.

## Figures and Tables

**Figure 1 vaccines-08-00481-f001:**
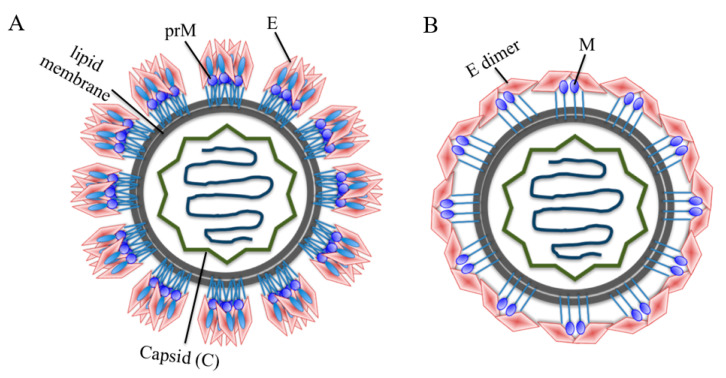
Schematic model of a flavivirus particle. (**A**) Immature flavivirus particles. The surface of immature particles consists of 60 trimeric spikes and each spike is composed of a prM-E heterodimer. (**B**) Mature flavivirus particles. Mature particles are formed after prM cleavage and each particle contains 90 E homodimers and 90 M homodimers. The E glycoproteins are involved in receptor binding, attachment and virus-cell fusion.

**Figure 2 vaccines-08-00481-f002:**
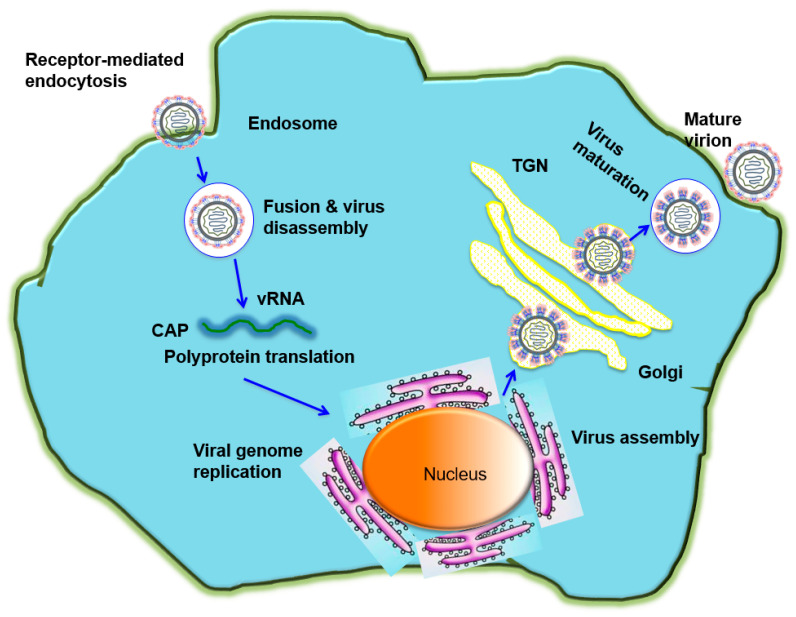
The flavivirus life cycle. A flavivirus attaches to the cell through its binding to a receptor on cell surface of a host cell and subsequently enters the cell by receptor-mediated endocytosis. The low pH environment in the endosome triggers conformational changes of E protein, resulting in the fusion between viral and endosomal membranes and release of the viral genome into the cytoplasm for replication. The immature viral and subviral particles are assembled on the surface of the endoplasmic reticulum (ER), transported through the trans-Golgi network (TGN), and then cleaved by the host protease furin, resulting in the formation of mature viral particles, which are subsequently released by exocytosis.

**Table 1 vaccines-08-00481-t001:** VLP-based ZIKV vaccines.

Expression System	VLP Components	Animal Model	Virus Strain	Immune Response	Clinical Trials	Ref.
Baculovirus-insect cells	prM-E	BALB/c mice	SZ-WIV01	Stimulated high levels of virus neutralizing antibody titers, ZIKV-specific IgG titers and potent memory T cell responses	No	[40]
HEK293 cells	prM-E	CB6F1 and AG129 mice	African MR766 and Brazilian SPH2015	Induced high ZIKV-specific neutralizing antibody titers and protected mice against weight loss, morbidity and mortality	No	[59]
HEK293 cells	prM-E	Interferon deficient AG129 mice and BALB/c mice	H/PF/2013	Induced neutralizing antibodies	No	[60]
293T cells	C-prM-E and prM-E	BALB/c mice	PRVABC59	Both prM-E- and C-prM-E-based VLP vaccines were highly effective in generating neutralizing antibodies, with the latter being more potent	No	[63]
Nicotiana benthamiana plants	E protein domain III (zDIII)	C57BL/6 mice	PRVABC59	Evoked potent neutralizing antibody and cellular immune responses; Circumvented induction of antibodies with ADE activity for DENV infection	No	[65]
Yeast *Pichia pastoris*	DIII	BALB/c mice and AG129 mice	SPH2015	Induced ZIKV-specific neutralizing antibodies, with no discernible ADE capacity	No	[64]

**Table 2 vaccines-08-00481-t002:** VLP-based DENV vaccines.

Expression System	VLP Components	Animal Model	Virus Strain	Immune Response	Clinical Trials	Ref.
Yeast *Pichia pastoris*	C-prM-E	rabbits	Singapore strain S275/90	Induced neutralizing antibodies	No	[67]
Expi293TM cells	C-prM-E	BALB/c mice	DENV-2 Th-36	Mice immunized with the VLP vaccine produced at 31 °C elicited the highest titer of neutralizing antibodies when compared to those elicited by equivalent doses of the vaccine produced at 37 °C	No	[68]
Baculovirus-insect cells	prM-E	BALB/c mice	SZ-WIV01	Stimulated high levels of virus neutralizing antibody titers, ZIKV-specific IgG titers, and potent memory T cell responses	No	[40]
Yeast *Pichia pastoris*	prM-E	BALB/c mice	DENV-1 GZ01/95	Induced neutralizing antibodies and T cell responses	No	[69]
293T cells	prM-E	BALB/c mice	DENV1 GZ01/95, DENV2 ZS01/01, DENV3 H87, DENV4 H241	Induced VLP-specific IgG and neutralizing antibodies, as well as cellular immune responses	No	[71]
HEK293 cells	prM-E	No test	DENV1 WestPac-74, DENV2 S16803, DENV3 CH53489, DENV4 TVP-376	No test	No	[48]
Mosquito cells	prM-E	BALB/c mice and cynomolgus macaques	DENV2 03-0420 and NS1-123	With the help of adjuvants, the VLPs induced strong virus neutralizing antibodies.	No	[73]
Yeast *Pichia pastoris*	prM-E	BALB/c mice	DENV1 GZ01/95, DENV2 ZS01/01, DENV3 H87, DENV4 H241	The tetravalent VLPs induced humoral and cellular immune responses against all four dengue virus serotypes. The antibody levels were higher with tetravalent than with monovalent immunization	No	[52]
FreeStyle 293F cells	prM-E with an F108 mutation in the fusion loop of E	BALB/c mice	DENV1, Western Pacific strain, DENV2, S1 vaccine strain, DENV3, Singapore 8120/95 strain, DENV4, ThD4_0476_97 strain	The tetravalent vaccine elicited potent neutralizing antibody responses against all four DENV serotypes, with ADE not observed against any serotype at a 1:10 serum dilution	No	[50]
Yeast *Pichia pastoris*	E	BALB/c and AG129 mice	DENV2 New Guinea	Induced potent DENV-2-specific neutralizing antibodies and protected AG129 mice against lethal challenge with a virulent DENV-2 strain	No	[72]
Yeast *Pichia pastoris*	E	BALB/c mice	DENV1 WP74, DENV4 TVP-360, DENV3 CH-53489	Elicited EDIII-specific neutralizing antibody response in mice	No	[70,78,81]
Yeast *Pichia pastoris*	EDIII	BALB/c mice	DENV2 (unmentioned strain)	Elicited DENV-2-specific antibodies which could neutralize its infectivity	No	[79]
*E. coli*	EDIII	BALB/c mice	DENV2 (unmentioned strain)	Induced specific antibodies capable of binding and neutralizing the infectivity of DENV2	No	[80]
Yeast *Pichia pastoris*	E protein of DENV1 and DENV2	BALB/c and AG129 mice	DENV1 WP74 and DENV2 New Guinea	The bivalent mosaic VLPs preserved the serotype-specific antigenic integrity of its two component proteins and elicited EDIII-specific neutralizing antibodies which lacked discernible ADE potential	No	[82]
Yeast *Pichia pastoris*	E protein of DENV1, DENV2, DENV3, DENV4	BALB/c and AG129 mice	DENV1 West Pac-74, DENV2 S16803, DENV3 CH53489, DENV4 TVP-360	The tetravalent VLPs were highly immunogenic and elicited EDIII-specific neutralizing antibodies against all four DENV serotypes, and they did not promote ADE *in vivo*	No	[83]
Yeast *Pichia pastoris*	EDIII of DENV1, DENV2, DENV3, DENV4	BALB/c, AG129, C57BL-6, C3H and Macaques	DENV1 West Pac-74, DENV2 PR159-S1/69, DENV3 H87/56, DENV4 H241-P	The tetravalent VLPs elicited serotype-specific neutralizing antibodies effective against all four DENV serotypes with negligible ADE potential	No	[86]

**Table 3 vaccines-08-00481-t003:** VLP-based WNV vaccines.

Expression System	VLP Components	Animal Model	Virus Strain	Immune Response	Clinical Trials	Ref.
*E. Coli* BL21DE3	DIII	BALB/c mice	WNV NY99	Three injections of the vaccine induced high titers of virus-neutralizing antibodies and completely protected mice from WNV infection	No	[94]
Baculovirus-insect cells	DIII	BALB/c mice	WNV Kunjin	Although relatively modest, DENV- and WNV-specific neutralizing antibodies were induced	No	[93]
*E. coli*	DIII	BALB/c mice	F101 New York	The VLP vaccine induced a broad spectrum of DIII-specific IgM, IgG1 and IgG2a/b antibodies	No	[95]
Herpes simplex virus-1 d106 recombinant virus-Vero cells	prM-E	BALB/c mice	unknown	The VLP vaccine induced a specific anti-WNV IgG antibody response in immunized mice	No	[55]
Baculovirus-insect cells	C, M and E	Not used	NY99034 EDV	Not tested	No	[96]

**Table 4 vaccines-08-00481-t004:** VLP-based JEV vaccines.

Expression System	VLP Components	Animal Model	Virus Strain	Immune Response	Clinical Trials	Ref.
J12#26 stable cell line	E	BALB/c mice	Beijing-1	Induced specific humoral and cell-mediated responses which were enhanced by the addition of adjuvants	No	[109]
Silkworm pupae	E	Mice and rabbits	Muar	Immunized mice and rabbit antisera showed plaque inhibition potency against homologous Muar and heterologous Nakayama, but less potency to Beijing-1 strain. Mixed immune rabbit antisera led to an increase in the antibody reaction to Beijing-1 strain	No	[111]
Silkworm Bm-N cells	prM-E	Mice and rabbits	Nakayma	Induced virus-neutralizing antibodies against Nakayama, Beijing-1, and Muar strains	No	[110]
CHO-heparan sulfate-deficient cell derived stable cell line	prM-E	Mice and swine	GI YL2009-4, GI TC2009-1, GIII SA14-14-2, GIII T1P1, GIII CH1392, GIII CJN, CII JKT1749, GIV JKT7003	GI-VLP-immunized mice and swine produced cross-protected antibody against GI and GIII JEVs	No	[112]
BJ-ME BHK21 stable cell line	prM-E	BALB/c mice	SA14-14-2	Elicited high titers of neutralizing antibodies which conferred 100% protection against lethal JEV challenge	No	[104]
Baculovirus-insect cells	prM-E	BALB/c mice	JEV-P3	Performed complete protection against viral challenge and significantly relieved pathological changes in mouse brain	No	[107]

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
