# Peer review of "Recent Advances in the Development of Virus-Like Particle-Based Flavivirus Vaccines"

_vaccines, 2020, doi:10.3390/vaccines8030481_

Round 1

Reviewer 1 Report

Here authors reviewed the current attempts to design vaccines against different flavivirus. The authors primarily focused on vaccine candidates based on virus-like particles (VLPs). However, they also covered attempts to design vaccines based on other modalities to elicit a viral immune response.

Overall, the manuscript is well written. Tables 1-4 are very helpful for readers to understand the current status of vaccine development against different flavivirus.

Please expand the abbreviation of YEV for the first time use in the text of the manuscript (page 3, line 64).

The authors may expand the shortcomings of Dengvaxia with current clinical trials.

Author Response

Here authors reviewed the current attempts to design vaccines against different flavivirus. The authors primarily focused on vaccine candidates based on virus-like particles (VLPs). However, they also covered attempts to design vaccines based on other modalities to elicit a viral immune response.

Overall, the manuscript is well written. Tables 1-4 are very helpful for readers to understand the current status of vaccine development against different flavivirus.

Please expand the abbreviation of YEV for the first time use in the text of the manuscript (page 3, line 64).

Response: Thanks. We have given the full name of yellow fever virus (YEV) when it is first used in paragraph 1 (line 28) to address Reviewer 2’s comment.

The authors may expand the shortcomings of Dengvaxia with current clinical trials.

Response: Thanks for the comments. We have added the following sentences in the revised manuscript to address these comments.

“Dengvaxia, a chimeric tetravalent dengue vaccine containing a live-attenuated YFV strain 17D backbone and prM and E genes of DENV1-4 serotypes, is unable to confer a balanced protection against all the serotypes. In addition, because of its low efficacy in children younger than nine years old, this vaccine enhances dengue infection in these individuals, increasing the risk of hospitalization and thus raising safety concerns” (lines 166-170).

Reviewer 2 Report

Vaccines – 873618 – review

This article is an extensive review on the use of VLPs in the field of Flavivirus vaccines and diagnostics. The authors review the VLP studies for the design of vaccines against Dengue virus, West-Nile Virus and Japanese Encephalitis Virus. The article has the merit of an extensive review of the literature on this topic. The authors could have improved the overall structure of the article. In some paragraphs, they mix different types of VLP technology. Since the authors have presented the vaccine VLP candidates in a table, there is no need to repeat the results of the mice studies in the text. Instead, the text could be more structured around the type of antigen expressed for the VLP (E, preM-E, C-preM-E, EDIII, …), the type of VLP technology (flavivirus VLP or HBc VLP, antigen design (modification of E), with the pro’s and con’s of the different approaches. Most of the information is present in the current version but not presented in a structured way.

Line 24-25 : Yellow-fever virus (YFV) should be mentioned in this paragraph. Even if there is a very effective vaccine, it is a major human pathogen.

Line 43 : a point is missing after (preM). Then, “on immature particles” instead of “on an immature particles”.

Line 63: it might be fair to mention veterinary vaccines against WNV for horses. They use different technologies: inactivated vaccines, YFV chimera, canarypox-vectored vaccine, DNA vaccine.

Line 72-73: YFV attenuated vaccine is recognized as a highly efficacious and safe vaccine. Should be mentioned.

Line 73-74: integration of plasmidic vaccine DNA into host genome is no more considered as a major risk. Several DNA vaccines are in clinical trials in humans to prevent infectious diseases or treat cancers. Several phase I and II trials are ongoing with DNA vaccines against Zika virus.

Line 76-77: the reason for the pre-existing immunity against adenovirus is not the use of adenovirus vectored vaccines but the natural infection with human adenovirus.

Introduction: Considering the existing pipeline of Zika vaccines in phase I or II, vaccine candidates are DNA vaccines, inactivated purified vaccines, live attenuated or vectored vaccines. As a general comment, it would be preferable to insist on the possible advantages of VLPs instead of the disadvantages of the other technologies, which, as they are listed, are not very relevant.

Paragraph 108-123: the authors mix in the same paragraph two different approaches: a/ VLPs using prM-E and C-prM-E constructs, b/ VLPs based on HBcAg which is a completely different type of VLP. The two technologies should be clearly separated and not included in the same paragraph for the sake of clarity. The HBc VLP technology is closer to the bacteriophage VLP technology (paragraph 124-130).

Paragraph 325-334: same remark on the mixing of different types of VLP technology in the same paragraph is confusing.

Paragraph 346-349: the authors should comment on the importance of the N-glycosylation site of preM and E proteins of JEV and its impact on the selection of the appropriate expression platform. Glycosylation is not the same depending on the expression platform (insect cell, human cell, yeast, …).

Author Response

Comments and Suggestions for Authors

This article is an extensive review on the use of VLPs in the field of Flavivirus vaccines and diagnostics. The authors review the VLP studies for the design of vaccines against Dengue virus, West-Nile Virus and Japanese Encephalitis Virus. The article has the merit of an extensive review of the literature on this topic. The authors could have improved the overall structure of the article. In some paragraphs, they mix different types of VLP technology. Since the authors have presented the vaccine VLP candidates in a table, there is no need to repeat the results of the mice studies in the text. Instead, the text could be more structured around the type of antigen expressed for the VLP (E, preM-E, C-preM-E, EDIII, …), the type of VLP technology (flavivirus VLP or HBc VLP, antigen design (modification of E), with the pro’s and con’s of the different approaches. Most of the information is present in the current version but not presented in a structured way.

Response: Thanks for the constructive comments to revise our manuscript. Based on these comments, we have made related revisions throughout the manuscript.

Line 24-25 : Yellow-fever virus (YFV) should be mentioned in this paragraph. Even if there is a very effective vaccine, it is a major human pathogen.

Response: Thanks. As suggested, yellow fever virus (YFV) has been added in paragraph 1 (line 28).

Line 43 : a point is missing after (preM). Then, “on immature particles” instead of “on an immature particles”.

Response: Thanks for carefully reading our manuscript. We have added a period “.” after “(prM)”, and corrected immature particles (lines 44 of the revised manuscript).

Line 63: it might be fair to mention veterinary vaccines against WNV for horses. They use different technologies: inactivated vaccines, YFV chimera, canarypox-vectored vaccine, DNA vaccine.

Response: As suggested, vaccines against WNV and those based on YFV chimera have been described (lines 70-73), and vaccines based on canarypox vector have been added in the revised manuscript (line 77).

Line 72-73: YFV attenuated vaccine is recognized as a highly efficacious and safe vaccine. Should be mentioned.

Response: Thanks for the comment. We have added one reference (Collins ND., et al. Curr Infect Dis Rep. 2017;19(3):14) and the following sentence to address this comment. “…live-attenuated yellow fever 17D vaccine is considered an effective and safe vaccine.…” (line 74 of the revised manuscript).

Line 73-74: integration of plasmidic vaccine DNA into host genome is no more considered as a major risk. Several DNA vaccines are in clinical trials in humans to prevent infectious diseases or treat cancers. Several phase I and II trials are ongoing with DNA vaccines against Zika virus.

Response: Thanks for the informative comment to improve our manuscript. We have removed this disadvantage of DNA vaccines, and changed this sentence as follows: “DNA-based vaccines encode viral-specific genes, and they are convenient for sequence editing; several DNA vaccines such as those targeting ZIKV have been proceeded in clinical trials ….” (lines 75-76 of the revised manuscript).

Line 76-77: the reason for the pre-existing immunity against adenovirus is not the use of adenovirus vectored vaccines but the natural infection with human adenovirus.

Response: Thanks for the comment. We have added the following sentences in the revised manuscript to address this comment: “Viral vectors, such as adenovirus and canarypox, can be used to develop vaccines against flaviviruses. A concern might be raised regarding pre-existing immunity against the viral vector (such as human adenovirus 5 (Ad5), which may result in reduced immunity and efficacy of the vaccines.” (lines 77-80).

Introduction: Considering the existing pipeline of Zika vaccines in phase I or II, vaccine candidates are DNA vaccines, inactivated purified vaccines, live attenuated or vectored vaccines. As a general comment, it would be preferable to insist on the possible advantages of VLPs instead of the disadvantages of the other technologies, which, as they are listed, are not very relevant.

Response: Thanks for the comment. We have revised the sentences in Introduction section as suggested (lines 81-89).

Paragraph 108-123: the authors mix in the same paragraph two different approaches: a/ VLPs using prM-E and C-prM-E constructs, b/ VLPs based on HBcAg which is a completely different type of VLP. The two technologies should be clearly separated and not included in the same paragraph for the sake of clarity. The HBc VLP technology is closer to the bacteriophage VLP technology (paragraph 124-130).

Response: Thanks for the comment to improve our manuscript. To clarify this, we have separated different categories of VLP-based ZIKV vaccines into two paragraphs, and further revised the related sentences (lines 110-130 of the revised manuscript).

Paragraph 325-334: same remark on the mixing of different types of VLP technology in the same paragraph is confusing.

Response: Thanks. We have modified these paragraphs, and added a title in each paragraph to describe the contents in each paragraph to avoid confusion (lines 284-307 of the revised manuscript).

Paragraph 346-349: the authors should comment on the importance of the N-glycosylation site of preM and E proteins of JEV and its impact on the selection of the appropriate expression platform. Glycosylation is not the same depending on the expression platform (insect cell, human cell, yeast, …).

Response: To address this comment, we have expanded the paragraph describing the factors affecting JEV VLP production, and indicated the difference of variant expression systems in production of effective VLPs with appropriate glycosylation (lines 334-342 of the revised manuscript).